# ALIGNSEP: TEMPORALLY-ALIGNED VIDEO-QUERIED SOUND SEPARATION WITH FLOW MATCHING

**Xize Cheng**[*], **Chenyuhao Wen**[*], **Tianhao Wang**[*], **Yongqi Wang**
**Zehan Wang**, **Rongjie Huang**, **Tao Jin**, **Zhou Zhao**
Zhejiang University[1], Independent Author[2]

## ABSTRACT

Video Query Sound Separation (VQSS) aims to isolate target sounds conditioned on visual queries while suppressing off-screen interference—a task central to audiovisual understanding. However, existing methods often fail under conditions of homogeneous interference and overlapping soundtracks, due to limited temporal modeling and weak audiovisual alignment. We propose **AlignSep**, the first generative VQSS model based on flow matching, designed to address common issues such as spectral holes and incomplete separation. To better capture cross-modal correspondence, we introduce a series of temporal consistency mechanisms that guide the vector field estimator toward learning robust audiovisual alignment, enabling accurate and resilient separation in complex scenes. As a *multi-conditioned generation* task, VQSS presents unique challenges that differ fundamentally from traditional flow matching setups. We provide an in-depth analysis of these differences and their implications for generative modeling. To systematically evaluate performance under realistic and difficult conditions, we further construct **VGGSound-Hard**, a challenging benchmark composed entirely of separation cases with homogeneous interference and strong reliance on temporal visual cues. Extensive experiments across multiple benchmarks demonstrate that AlignSep achieves state-of-the-art performance both quantitatively and perceptually, validating its practical value for real-world applications. More results and audio examples are available at: `https://AlignSep.github.io`.

## 1 INTRODUCTION

Video-Queried Sound Separation (VQSS) (Tzinis et al., 2022; Chen et al., 2023; Dong et al., 2022; Cheng et al., 2024) aims to isolate sound sources that correspond to visual content in a video while suppressing off-screen interference. As a core task in audio-visual understanding, VQSS facilitates applications such as video editing, accessibility enhancement, and content analysis. Early work in sound separation (Wang & Chen, 2018; Pegg et al., 2023) focused primarily on speech, addressing the classic *cocktail party problem*—separating clean speech from noisy environments. While traditional unsupervised methods (Roweis, 2000; Kristjansson et al., 2006; Hershey et al., 2016) advanced speech separation, they often struggled with interference from sources with similar timbres. To overcome this, researchers introduced visual speech enhancement (Gao & Grauman, 2021; Hsu et al., 2023; Lei et al., 2024), enabling more accurate separation in visually grounded contexts.

As the field evolved, research expanded beyond speech to more diverse sound sources in complex real-world environments (Kavalerov et al., 2019; Liu et al., 2023b), including instrument separation (Défossez et al., 2019; Luo & Yu, 2023) and general-purpose audio disentanglement (Wisdom et al., 2020). To increase flexibility, multimodal query approaches emerged (Liu et al., 2024; Ma et al., 2024), using text to describe target sounds. However, text descriptions often fail to fully capture the nuances of real-world sound-producing objects and events. This limitation has led to a resurgence of interest in video-guided sound separation, where visual information provides a richer and more precise grounding for identifying target audio sources.

Despite these advances, current video-guided sound separation methods still face two key limitations (Figure 1): **(1) Lack of Temporal Modeling.** Dong et al. (2022); Cheng et al. (2024) rely heavily on

---
[*]Equal Contribution

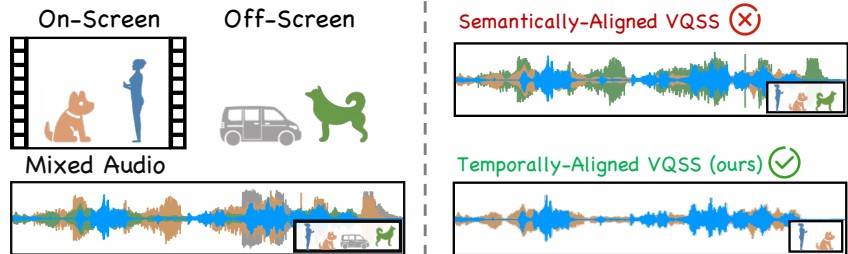

Figure 1: **Comparison of different video-queried sound separation methods.** Traditional category-based methods rely solely on semantic information, struggling to distinguish same-source sounds across on/off-screen regions (e.g., multiple dogs barking). In contrast, our proposed AlignSep can capture audiovisual consistency, enabling precise separation of on/off-screen sounds.

semantic cues, which are insufficient for separating acoustically similar sources. For instance, distinguishing multiple barking dogs across on-screen and off-screen regions requires capturing temporal alignment between visual motions and corresponding audio energy, not just semantic categories. **(2) Limitations of Mask-Based Methods.** The dominant separation paradigm uses time–frequency masking (Liu et al., 2023b; Chen et al., 2023), which struggles with overlapping soundtracks. In scenarios where multiple sources overlap in both time and frequency, mask-based methods fail to recover clean and distinct signals (Yuan et al., 2024), leading to incomplete separation or artifacts.

To address the aforementioned challenges, we propose **AlignSep**—the first generative temporal-aligned video-queried sound separation model based on flow matching (Lipman et al., 2022) designed for robust audiovisual separation. Unlike prior generative separation models (Yuan et al., 2024; Wang et al., 2024a), VQSS requires maintaining precise temporal alignment between audio and visual streams. To this end, we design a dedicated vector field estimator that explicitly preserves audio-visual alignment by employing a simple yet effective temporal concatenation strategy, combined with a cross-attention-free feed-forward Transformer encoder to enforce temporal consistency across modalities. Furthermore, unlike traditional flow-matching tasks—which are typically single-conditioned (e.g., text-to-audio (Huang et al., 2023; Liu et al., 2023a), text-to-image (Zhang et al., 2023) and video-to-audio (Wang et al., 2024b))—VQSS is a multi-conditioned generation task, where both the raw noisy audio and the video sequence jointly condition the output. This fundamental difference leads to new challenges in vector field learning and transport path modeling. We provide an in-depth analysis of these differences, and explain why straightforward rectification and deterministic acceleration techniques (e.g., in rectified flow) are suboptimal in this multi-conditioned setting. We further discuss the trade-offs between performance and efficiency specific to sound separation tasks under these constraints. To facilitate rigorous evaluation of VQSS models under realistic and difficult conditions, we construct **VGGSound-Hard**, a novel benchmark composed entirely of separation cases involving homogeneous interference and strong reliance on fine-grained temporal visual cues. Experimental results on multiple benchmarks—including MUSIC-Clean (Dong et al., 2022), VGGSound-Clean (Dong et al., 2022), and the proposed VGGSound-Hard—demonstrate the effectiveness of AlignSep, highlighting its potential as a next-generation framework for video-guided sound separation. Our main contributions are as follows:

- We revisit the task of video-queried sound separation (VQSS) and provide a detailed analysis of its unique challenges, including homogeneous interference, overlapping soundtracks, and the need for precise audio-visual temporal alignment.

- We propose **AlignSep**, a novel generative temporal-aligned VQSS framework based on conditional flow matching, designed to robustly model multi-conditioned generation by leveraging temporal visual cues and preserving cross-modal consistency.

- We introduce **VGGSound-Hard**, a new benchmark specifically curated to evaluate temporal alignment under real-world homogeneous interference, consisting of co-occurring on-/off-screen same-category sound sources.

- Extensive experiments on three benchmarks—MUSIC-Clean, VGGSound-Clean, and VGGSound-Hard—demonstrate that AlignSep achieves state-of-the-art performance in both quantitative metrics and human perceptual scores (e.g., MOS), validating its effectiveness in real-world audiovisual separation tasks.

## 2 RELATED WORKS

### 2.1 UNIVERSAL SOUND SEPARATION

Universal sound separation (Kavalerov et al., 2019; Liu et al., 2022; Pons et al., 2024) aims to extract distinct audio tracks from mixed signals, serving as a fundamental task for audio understanding. Early research primarily focused on domain-specific separation, particularly in speech (Li et al., 2023; Pegg et al., 2023; Wang et al., 2023; Li et al., 2022) and music (Défossez et al., 2019; Manilow et al., 2022; Rouard et al., 2023; Luo & Yu, 2023) separation. Subsequent work (Kavalerov et al., 2019) introduced Permutation Invariant Training (Yu et al., 2017) to separate mixed audio into multiple unidentified categories. However, these methods are largely limited to music, speech, and artificial sounds, struggling with complex real-world auditory scenes. To address this, researchers have developed large-scale annotated audio datasets (Gemmeke et al., 2017; Chen et al., 2020), advancing universal sound separation. Recent approaches (Ochiai et al., 2020; Kong et al., 2020; Liu et al., 2022) leverage class labels as queries for targeted sound separation. Nevertheless, textual descriptions inherently lack the capacity to fully characterize real-world auditory information, prompting exploration of visual-queried sound sepration. We revisit the task of video-queried sound separation (VQSS) and provide a detailed analysis of its unique challenges, including homogeneous interference, overlapping soundtracks, and the need for precise audio-visual temporal alignment.

### 2.2 VISUAL-QUERIED SOUND SEPARATION

In the field of visual-queried sound separation, AudioScope 1&2 (Tzinis et al., 2020; 2022) introduce a novel joint audio-visual classifier to identify object categories that appear in the video and produce the corresponding sounds in the audio track. Building upon this foundation, i-Query (Chen et al., 2023) employs an advanced cross-attention mechanism to detect sound-emitting objects within video sequences, though it relies on pre-extracted object bounding boxes as input. Recent advancements (Dong et al., 2022; Cheng et al., 2024) leveraged powerful visual pre-trained models to extract rich semantic information from visual content. However, current VQSS methods (Tzinis et al., 2022; Chen et al., 2023; Dong et al., 2022; Cheng et al., 2024) focus heavily on spatial features but neglect audio-visual temporal alignment, leading to ambiguity in distinguishing sounds from on-screen versus off-screen objects of the same category. To address this challenge, we introduce VGGSound-Hard, a new benchmark specifically curated to evaluate temporal alignment under real-world homogeneous interference, consisting of co-occurring on-/off-screen same-category sound sources.

### 2.3 GENERATIVE SOUND SEPARATION

Traditional sound separation methods (Dong et al., 2022; Cheng et al., 2024) often struggle with overlapping sound events, as mask-based discriminative models may produce spectral holes (Wang et al., 2022), limiting their effectiveness in complex acoustic environments. To address this, researchers have turned to non-discriminative models, initially leveraging Generative Adversarial Networks (GANs) (Chen et al., 2024) to improve perceptual quality. Subsequent work has explored diffusion models (Hai et al., 2024) and flow-matching (Yuan et al., 2024) for more natural and refined separation. Despite these advancements, generative visual-queried sound separation remains unexplored. Inspired by vision-to-audio generation (Wang et al., 2025), we propose AlignSep, a novel generative VQSS framework based on conditional flow matching, designed to robustly model multi-conditioned generation by leveraging temporal visual cues and preserving cross-modal consistency.

## 3 ALIGNSEP

### 3.1 OVERVIEW

Let a mixed audio signal $A^m = \{A_1^m, \cdots, A_n^m\}$ be a linear superposition of a clean audio source $A^c = \{A_1^c, \cdots, A_n^c\}$ and an interfering audio source $A^i = \{A_1^i, \cdots, A_n^i\}$. Video-Queried Sound Separation (VQSS) aims to extract a clean audio signal $A^c$ that maintains strict temporal alignment with the corresponding visual frame sequence $V = \{V_1, \cdots, V_k\}$, where $n$ and $k$ denote the number

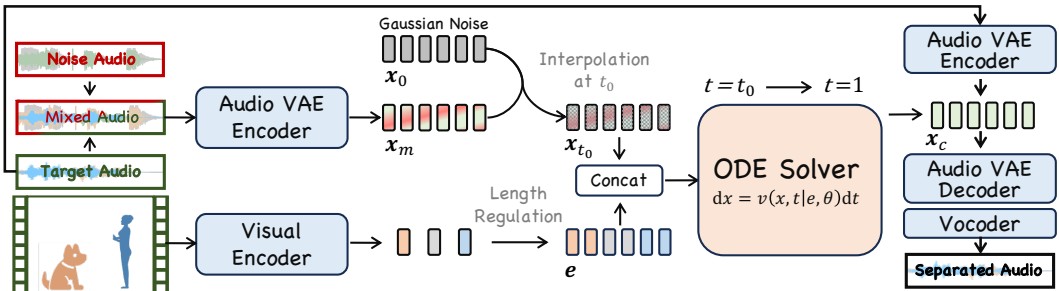

Figure 2: llustration of AlignSep. AlignSep is a video-queried sound separation model based on flow-matching, designed to establish a mapping from the distribution of mixed audio and the distribution of separated audio, conditioned on visual information. Given a mixed audio input, we first perturb it with Gaussian noise and then progressively denoises it, guided by the visual condition **c**. This process gradually transforms the mixed noisy audio into separated audio that is temporally aligned with the driving video.

of audio frames and video frames, respectively. As illustrated in Figure 2, we introduce AlignSep, a conditional flow-matching framework designed for precisely temporal-aligned video-queried sound separation. In Section 3.2, we provide a detailed introduction of the flow-matching-based sound separation model. Section 3.3 elaborates on the architectural details of AlignSep, while Sections 3.4 and 3.5 present specialized training strategies aimed at enhancing cross-modal alignment capabilities.

## 3.2 FLOW-MATCHING BASED SOUND SEPARATION

Conditional sound separation tasks, such as TQSS (Liu et al., 2022) and IQSS (Dong et al., 2022), can be formulated as a conditional mapping from the distribution of mixed audio, $\boldsymbol{x}_m \sim p_m(\boldsymbol{x})$, to the distribution of clean audio, $\boldsymbol{x}_c \sim p_c(\boldsymbol{x})$. This process can be interpreted as a time-dependent probability density transformation (i.e., flow), governed by the following ordinary differential equation (ODE):

$$\mathrm{d}\boldsymbol{x} = \boldsymbol{u}(\boldsymbol{x}, t, \boldsymbol{e})\mathrm{d}t, \quad t \in [0, 1] \tag{1}$$

where $t$ represents the time position, $\boldsymbol{x}$ is a point in the probability density space at time $t$, $\boldsymbol{u}$ denotes the transport vector field (i.e., the probability gradient with respect to $t$), and $\boldsymbol{e}$ is the conditioning embedding. In Video-Queried Sound Separation (VQSS), the conditioning signal $\boldsymbol{e}$ corresponds to visual features extracted from video frames. The source distribution $\boldsymbol{x}_m$ and target distribution $\boldsymbol{x}_c$ represent the compressed mel-spectrogram latents of the mixed audio $A^m$ and clean audio $A^c$, respectively, obtained via a pre-trained VAE encoder (Liu et al., 2023a).

**Conditional Flow Matching Model.** The fundamental principle of flow-matching generative model is to train a neural network $\theta$ to approximate the transport vector field $\boldsymbol{u}$ with the flow-matching objective:

$$L_{\mathrm{FM}}(\theta) = \mathbb{E}_{t, p_t(\boldsymbol{x})}\big\|v(\boldsymbol{x}, t, \boldsymbol{e}; \theta) - u(\boldsymbol{x}, t, \boldsymbol{e})\big\|^2, \tag{2}$$

where $p_t(\boldsymbol{x})$ denotes the distribution of $\boldsymbol{x}$ at timestep $t$. However, in practice, directly computing $\boldsymbol{u}(\boldsymbol{x}, t, \boldsymbol{e})$ is intractable due to the lack of explicit knowledge of the target distribution $p_c(\boldsymbol{x})$, as well as the unknown forms of $p_t(\boldsymbol{x})$ and $\boldsymbol{u}$. To circumvent this challenge, we adopt the conditional flow-matching (CFM) objective (Lipman et al., 2022), designing specific probabilistic paths that enable efficient sampling from $p_t(\boldsymbol{x} \mid \boldsymbol{x}_c)$ and facilitate the computation of $u(\boldsymbol{x}, \boldsymbol{e}, t \mid \boldsymbol{x}_c)$:

$$L_{\mathrm{CFM}}(\theta) = \mathbb{E}_{t, p_c(\boldsymbol{x}_c), p_t(\boldsymbol{x}, \boldsymbol{x}_c)}\big\|v(\boldsymbol{x}, t, \boldsymbol{e}; \theta) - u(\boldsymbol{x}, t, \boldsymbol{x}_c, \boldsymbol{e})\big\|^2. \tag{3}$$

**ODE Solving and Audio Reconstruction.** Once the latent field estimator $\theta$ is trained, we use numerical solvers to approximate the solution of the ODE $\mathrm{d}\boldsymbol{x} = v(\boldsymbol{x}, t, \boldsymbol{e}; \theta)$ at discretized time steps. A simple yet widely used solver is the Euler method, which updates $\boldsymbol{x}$ iteratively as follows:

$$\boldsymbol{x}_{t+\epsilon} = \boldsymbol{x} + \epsilon v(\boldsymbol{x}, t, \boldsymbol{e}; \theta) \tag{4}$$

where $\epsilon$ is the step size. Finally, the sampled latent representation is passed through the VAE decoder to reconstruct the mel-spectrogram, followed by a vocoder (Lee et al., 2022) to generate the final audio waveform.

### 3.3 Detailed Model Architecture

**Audio Encoder & Visual Encoder.** To achieve effective temporal encoding of video sequences, we employ the pre-trained temporal visual encoder (CAVP) from prior V2A work (Luo et al., 2023). Distinguished from the global-level representation like ImageBind (Han et al., 2023) that primarily focus on semantic content representation, CAVP incorporates temporal synchronization supervision between video and audio modalities. This enables the encoder to capture dynamic temporal correlations across video frames, rather than relying solely on static semantic features. As for audio modality, to align the distributions of mixed and clean audio as closely as possible, we use a pre-trained VAE audio encoder (Liu et al., 2023a) to map both into a shared audio latent space. During inference, the paired VAE decoder can then directly convert the latent features back into mel spectrograms.

**Temporally-Aligned Vector Field Estimator.** As the core component of our model, the vector field estimator employs a feedforward Transformer architecture. Given the critical importance of temporal alignment, we adopt a concatenation-based approach to effectively fuse multimodal features. Specifically, after extracting video features $e$, which has a dimension of 512, via CAVP and latent-space audio features $x_m$, which has a dimension of 20, via VAE, we first expand the video features to match the time dimension of the audio features, ensuring precise temporal correspondence. These aligned features are then concatenated, followed by the timestep-encoded vector $t$ appended at the end of the sequence. This structured input is subsequently fed into the main model for noise prediction, allowing it to leverage temporally coherent information for improved performance.

### 3.4 Classifier-Free Guidance

For our generative sound separation task, it can be considered a variant of a video-to-audio task. Therefore, we employ classifier-free guidance. This method effectively combines conditional and unconditional output, allowing us to strike a balance between quality and diversity. The sampling procedure for our audio generation with classifier-free guidance can be formulated as follows:

$$\hat{v}(x, t, e; \theta) = s \cdot v(x, t, e; \theta) + (1 - s) \cdot v(x, t, \emptyset; \theta) \tag{5}$$

Here, $s > 1$ represents the classifier sampling scale, which adjusts the balance between the diversity and quality of the generated samples. The ODE solver conditioned with $\emptyset$ is realized by randomly dropping the latent variable $e$ and replacing it with a "null" embedded representation. This latent input exchange facilitates the sampling procedure for video-to-audio generation, as we have established a modality-aligned latent space. We set $s$ to 4.5 in our experiments.

## 4 VGGSound-Hard: A Benchmark for Sound Separation under Homogeneous Real-World Interference

### 4.1 Homogeneous Audio Pair Construction

All samples in VGGSound-Hard are sourced from the VGGSound test set. To construct mixtures with **homogeneous real-world interference**, we first group audio clips by their category labels. Within each category, we compute pairwise cosine similarity using the CLAP audio encoder and select the highest-scoring pairs, yielding approximately 2,000 candidate homogeneous audio pairs. Following the synthesis pipeline of CLIPSep (Dong et al., 2022), each same-category pair is then mixed to produce a preliminary set of homogeneous mixed-audio samples.

### 4.2 Human Verification

To ensure that the final data reliably supports the evaluation of **temporal alignment in sound separation**, we perform an additional stage of manual verification. A trained audio–visual annotator reviews all candidate samples and applies the following criteria: (1) **Clear temporal cues in the video:** The visual channel must present actions with identifiable rhythmic or temporal structure, allowing annotators to infer the timing of corresponding sound events (e.g., excluding horn sounds without visible motion cues). (2) **On-screen sound sources:** All target sound events must be physically present in the associated video frames, ensuring that the model is not required to separate

Table 1: Comparison of visually-queried sound separation performance on MUSIC-Clean, VGGSound-Clean, and VGGSound-Hard (VG-Hard). The evaluation considers semantic consistency between audio–audio ($S_{A-A}$), semantic consistency between audio–visual ($S_{A-V}$), and temporal consistency between audio–visual ($T_{A-V}$) to assess the quality of the separated results. † Since Davis is originally trained on different datasets, we retrained their models on the same dataset to ensure a fair comparison.

| Method | Temp. Align | VGGSOUND-Clean | | | Music-Clean | | | VS-Hard |
|---|---|---|---|---|---|---|---|---|
| | | $S_{A-A}\uparrow$ | $S_{A-V}\uparrow$ | $T_{A-V}\uparrow$ | $S_{A-A}\uparrow$ | $S_{A-V}\uparrow$ | $T_{A-V}\uparrow$ | $T_{A-V}\uparrow$ |
| Target Audio | ✗ | 100.00 | 39.33 | 95.83 | 100.00 | 37.10 | 82.22 | 94.07 |
| Mixed Audio | ✗ | 63.20 | 19.71 | 61.46 | 52.96 | 15.18 | 28.89 | 73.73 |
| CLIPSEP (Dong et al., 2022) | ✗ | 66.74 | 24.21 | 79.17 | 60.59 | 21.42 | 51.11 | 85.59 |
| i-Query (Chen et al., 2023) | ✗ | 68.14 | 26.93 | 80.78 | 66.29 | 24.46 | 64.21 | 79.52 |
| OmniSep (Cheng et al., 2024) | ✗ | 70.83 | 27.57 | 81.25 | 67.67 | 25.74 | **68.89** | 76.27 |
| †Davis (Huang et al., 2024) | ✗ | 62.53 | 23.18 | 74.39 | 61.32 | 22.49 | 56.47 | 74.13 |
| †Davis-flow (Huang et al., 2025) | ✗ | 65.82 | 24.21 | 82.32 | 69.21 | 27.76 | 65.71 | 76.27 |
| AlignSep (ours) | ✓ | **73.38** | **27.89** | **96.88** | **72.28** | **28.92** | 66.67 | **95.76** |

Table 2: Mean Opinion Score (MOS) across four evaluation dimensions: Noise Residuals (NR), Audio-Visual Consistency (AVC), Audio Quality (AQ), and Overall Score (OS).

| Method | VGGSound-Clean | | | | Music-Clean | | | | VGGSound-Hard | | | |
|---|---|---|---|---|---|---|---|---|---|---|---|---|
| | NR | AVC | AQ | OA | NR | AVC | AQ | OA | NR | AVC | AQ | OA |
| ClipSep | 3.31 | 3.31 | 3.31 | 3.85 | 2.91 | 3.91 | 3.82 | 3.55 | 3.57 | 4.36 | **4.29** | 4.14 |
| OmniSep | 3.62 | 3.69 | 3.85 | 3.62 | **4.09** | 4.19 | 3.82 | 4.01 | 3.29 | 4.29 | 4.21 | 4.07 |
| AlignSep | **4.23** | **4.53** | **4.08** | **4.31** | 3.82 | **4.27** | **4.18** | **4.18** | **4.21** | **4.64** | 4.21 | **4.43** |

sounds originating off-screen. After this filtering process, we obtain 118 high-quality audio–visual pairs that exhibit strong semantic homogeneity yet clearly distinct temporal patterns, forming the final VGGSound-Hard benchmark.

## 5 EXPERIMENTS

### 5.1 BENCHMARKING VIDEO-QUERIED SOUND SEPARATION

Video-queried sound separation (VQSS) poses unique challenges and requires dedicated evaluation methodologies. Unlike conventional source separation, which primarily focuses on recovering semantically correct signals, VQSS additionally demands that the separated audio be temporally aligned with the visual stream. For instance, even if a separated sound is semantically consistent with the scene, it should not be preserved if it is not actually produced by the objects in the video. Existing benchmarks fail to capture this dual requirement. To address this limitation, we establish a new benchmark that introduces tailored evaluation metrics and dataset settings for VQSS.

**Evaluation Metrics.** Conventional source separation is usually assessed with reconstruction-based metrics such as SDR (Vincent et al., 2006), which correlate poorly with human perception (Le Roux et al., 2019; Cartwright et al., 2018). The gap is even larger for generative methods (Hai et al., 2024; Yuan et al., 2024), where minor waveform deviations can lead to large but perceptually irrelevant errors. We therefore adopt a comprehensive evaluation protocol covering two dimensions: (1) Semantic Alignment: CLAP for audio–audio (A–A) and ImageBind for audio–visual (A–V) consistency; (2) Temporal Synchronization: alignment accuracy (Acc) (Luo et al., 2023), following Video-to-Audio evaluation (Xing et al., 2024; Wang et al., 2025).

To bridge the gap between objective scores and human perception, we further conduct Mean Opinion Score (MOS) evaluations, where human raters assess four aspects of the generated outputs: Noise Residuals (NR), Audio-Visual Consistency (AVC), Audio Quality (AQ), and Overall Score (OS).

MOS provides complementary insights into perceptual quality and cross-modal coherence, ensuring a more faithful assessment of real-world performance. The detailed evaluation protocol and rating procedure are provided in the Appendix B.

**Datasets.** As an early effort in VQSS, Dong et al. (2022) curated VGGSOUND-Clean and MUSIC-Clean, where target and interference sounds are drawn from different categories, representing relatively simple separation cases. To better reflect real-world scenarios—where overlapping sounds often come from the same class and are harder to disentangle—we introduce VGGSOUND-Hard, a new benchmark in which both target and interference sounds are from the same category and the target audio must be highly temporally aligned with the visual stream. This provides a more realistic and challenging testbed for advancing VQSS research.

## 5.2 IMPLEMENTATION DETAILS

Following recent video-to-audio (V2A) works (Luo et al., 2023; Wang et al., 2025), we downsampled all audio signals into 16 kHz and converted them to Mel-spectrograms with 80 frequency bins and a hop size of 256. All videos were downsampled to 4 FPS. All data samples were truncated into 8-second clips for both training and inference. The transformer architecture in our vector field estimator comprises 4 layers with a hidden dimension of 576. For waveform synthesis, we used a pretrained BigVGAN vocoder (Lee et al., 2022). Additional11implementation details are provided in Appendix A.

## 5.3 MAIN RESULTS

Table 1 presents the quantitative results of visual-queried sound separation on MUSIC-Clean, VGGSound-Clean, and the more challenging VGGSound-Hard benchmark. Compared with the baselines, CLIPSep (Dong et al., 2022) and OmniSep (Cheng et al., 2024), our proposed **AlignSep** achieves consistent and notable improvements across all datasets. On MUSIC-Clean and VGGSound-Clean, AlignSep yields higher scores in both semantic consistency ($S_{A-A}$, $S_{A-V}$) and temporal alignment ($T_{A-V}$), demonstrating its capability to leverage both semantic and temporal cues. Particularly on the VGGSOUND-Clean benchmark, AlignSep reaches 96.88% in $T_{A-V}$, significantly surpassing the baselines, while on VGGSound-Hard it maintains a strong 95.76% despite the increased difficulty. In contrast, while OmniSep improves semantic consistency compared with CLIPSep, its performance on VGGSound-Hard drops to 76.27% $T_{A-V}$, suggesting limited capacity in modeling temporal relationships under complex acoustic-visual mixtures. These results collectively demonstrate that AlignSep not only strengthens semantic alignment but also effectively captures fine-grained temporal correspondence, leading to superior sound separation quality across both clean and challenging benchmarks.

To further assess perceptual quality, Table 2 reports Mean Opinion Scores (MOS) across four human evaluation dimensions: Noise Residuals (NR), Audio-Visual Consistency (AVC), Audio Quality (AQ), and Overall Score (OA). AlignSep consistently outperforms both CLIPSep and OmniSep across all datasets and evaluation aspects. On Music-Clean, it achieves the highest ratings in every dimension (e.g., 4.53 AVC, 4.31 OA), highlighting its effectiveness in producing clean, temporally aligned, and perceptually natural audio. On VGGSOUND-Clean, while OmniSep scores highest in NR (4.09), AlignSep leads in AVC (4.27), AQ (4.18), and OA (4.18), suggesting better perceived audio-visual coherence. For the most challenging VGGSound-Hard set, AlignSep again delivers the best overall score (4.43), showing robustness even under complex and ambiguous conditions. Together, these subjective results corroborate the quantitative gains, indicating that AlignSep not only improves alignment metrics but also translates into better human-perceived separation quality.

## 5.4 TEMPORALLY-ALIGNED VIDEO-QUERIED SOUND SEPARATION

To compare the temporal alignment ability of different VQSS methods, we evaluate their performance when varying the number of reference video frames used as queries. For OmniSep and CLIPSep, which do not inherently model temporal dynamics, we adapt them by segmenting the mixed audio according to each frame's semantic information, thereby enforcing temporal alignment. Figure 3 illustrates the impact of reference frame rate (FPS) on alignment accuracy across different

Table 3: Evaluation of different denoising steps on VGGSound-Clean, MUSIC-VGGSound, and VGGSound-Hard. We report ImageBind similarity, CLAPScore, and alignment accuracy (AlignAcc). The last two columns present inference time and corresponding throughput (FPS). Best results are highlighted in **bold**.

| Method | VGGSOUND-Clean | | | Music-Clean | | | VS-Hard | FPS↑ |
|---|---|---|---|---|---|---|---|---|
| | $S_{A-V}$↑ | $S_{A-A}$↑ | $T_{A-V}$↑ | $S_{A-V}$↑ | $S_{A-A}$↑ | $T_{A-V}$↑ | $T_{A-V}$↑ | |
| AlignSep(Step=5) | 25.11 | 64.47 | 85.42 | 22.13 | 62.17 | 44.44 | 88.14 | **5.56** |
| AlignSep(Step=10) | 26.66 | 69.86 | 92.71 | 24.92 | 68.51 | 53.33 | 94.07 | 4.00 |
| AlignSep(Step=25) | 27.89 | 73.38 | **96.88** | 28.92 | 72.28 | 66.67 | **95.76** | 2.17 |
| AlignSep(Step=50) | **28.12** | 73.50 | 95.83 | 30.70 | 72.13 | 68.89 | 93.22 | 1.35 |
| AlignSep(Step=100) | 2756 | **73.64** | **96.88** | **30.88** | **72.80** | **68.89** | 93.22 | 0.72 |
| OmniSep | 27.57 | 70.83 | 81.25 | 25.74 | 67.67 | 68.89 | 76.27 | 11.2 |
| AlignSep$_{\text{Rectified Flow}}^{\text{step=100}}$ | 21.39 | 57.36 | 84.38 | 21.41 | 49.76 | 46.67 | 92.37 | 0.77 |

methods. AlignSep shows a clear upward trend: as the FPS increases, its performance steadily improves and saturates at higher frame rates, rising from about 0.76 at 0.25 FPS to nearly 0.95 at 4 FPS.

These results demonstrate that AlignSep effectively leverages fine-grained temporal cues to enhance separation quality.

In contrast, ClipSep shows an almost flat curve around 0.81 across all frame rates. These image-based methods rely solely on semantic modeling and lack temporal information, resulting in weak dependency on visual temporal resolution and a clear limitation in exploiting alignment signals. OmniSep, while benefiting from enhanced semantic alignment that improves its performance on simpler datasets such as VGGSound-Clean, struggles on more challenging benchmarks like VGGSound-Hard that demand temporal modeling. Its emphasis on cross-modal semantic alignment further restricts the modeling of temporal correspondence, leading to even lower separation performance in such scenarios.

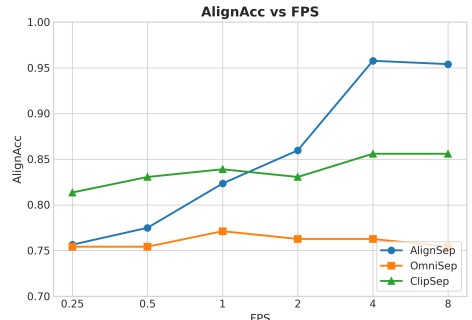

Figure 3: Comparison of sound separation performance with different levels of temporal information on VGGSOUND-Hard. OmniSep represents the performance when relying solely on semantic information. The x-axis indicates the number of video frames per second (FPS) used for VQSS.

## 5.5 PERFORMANCE AND EFFICIENCY DISCUSSION IN SOUND SEPARATION

**Generative methods meet the core requirements of VQSS.** Although generative sound separation methods are slightly slower than traditional mask-based approaches, their ability to directly generate waveforms makes them naturally suited for Video-Queried Sound Separation (VQSS). They effectively address two of the most critical challenges in this task: **(1) Disentangling overlapping signals.** The objective of VQSS is to isolate the target audio source from a mixture, guided by visual input. Generative models perform iterative inference with cross-modal conditioning at each step, allowing them to enforce mixture and phase consistency. This iterative refinement gradually routes ambiguous energy to the correct source and reduces leakage. As shown in the results, AlignSep's performance improves steadily with more denoising steps—on VGGSOUND-Clean, the audio–visual semantic score ($S_{A-V}$) increases from $64.47 \to 73.38 \to 73.64$ as the number of steps increases from $5 \to 25 \to 100$, while the temporal alignment score ($T_{A-V}$) reaches 96.88 at both 25 and 100 steps. In contrast, traditional mask-based methods struggle to separate sources with similar frequency bands and often suffer from severe separation artifacts, leading to degraded perceptual quality. **(2) Capturing fine-grained temporal alignment.** VQSS depends heavily on frame-level temporal cues, such as lip movements, object collisions, and transitions in and out of the frame. Traditional methods like OmniSep have limited capacity to model such fine temporal structure.

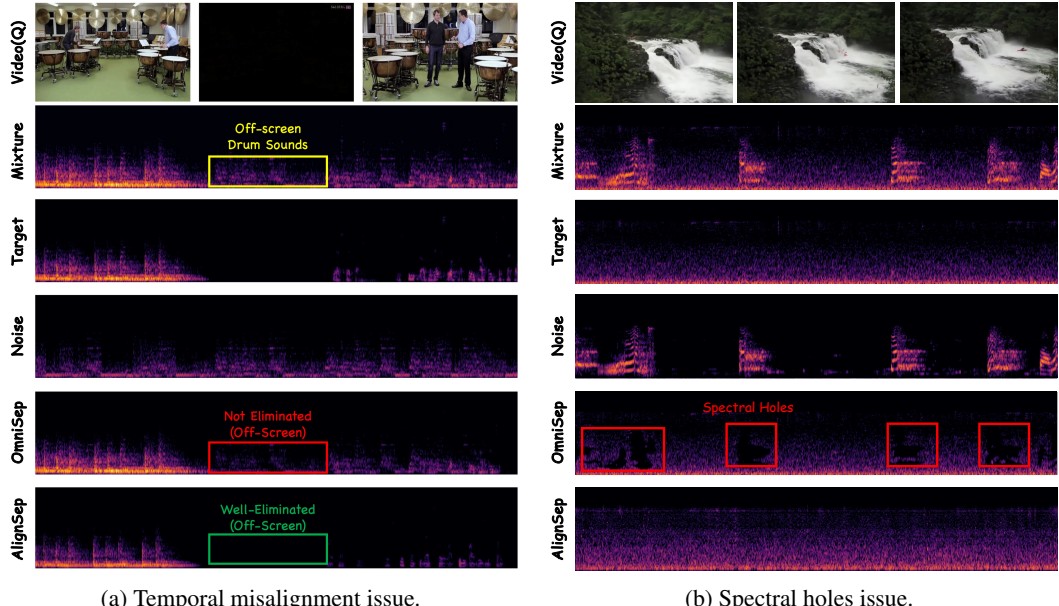

(a) Temporal misalignment issue.       (b) Spectral holes issue.

Figure 4: Qualitative comparison of VQSS. (a) illustrates a temporal misalignment case, while (b) demonstrates the spectral holes artifact. We highlight the critical regions using different colors.

In comparison, conditional generative models can explicitly incorporate temporal information and refine separation results accordingly across iterations. With schedulable guidance, they can progressively enhance temporal alignment. For example, on the VS-Hard benchmark, AlignSep achieves a high alignment score of $T_{A-V} = 95.76$ with just 25 inference steps.

**Traditional Rectified-flow acceleration struggles in VQSS.** While *Rectified Flow* (RF) achieves fast and deterministic sampling by straightening the generative trajectory into a smooth ODE path (Lipman et al., 2022), its effectiveness diminishes in complex multi-conditioned settings such as VQSS. In VQSS, the model is required to condition jointly on an audio mixture $m$ and a sequence of video frames $v_{1:T}$, and to perform time–frequency–object routing, i.e., assigning acoustic energy to the correct visual source over time.

This results in a posterior distribution $p(s \mid m, v_{1:T})$ that is highly multi-modal and piecewise non-smooth, often exhibiting discrete bifurcations and high-curvature transport paths. Under such structure, a single deterministic trajectory—as used in RF—tends to bias toward high-density regions and effectively averages across modes. Moreover, RF lacks the iterative correction mechanism found in diffusion models—specifically, the loop of denoising followed by consistency projection. This feedback loop is essential for correcting early assignment errors in multi-modal generation tasks. As a consequence, RF struggles to represent the complex conditional structure in VQSS, often leading to misalignment between generated results and the conditioning inputs. As shown in Table 3, this results in significantly lower performance: RF (with 100 sampling steps) achieves $S_{A-V} = 57.36$, markedly below the 73.64 achieved by the diffusion-based AlignSep model—despite comparable inference speeds under efficient samplers.

**VQSS requires fewer sampling steps than other generation tasks.** Unlike other conditional generation tasks (e.g., text-to-audio synthesis), **Visual-Query Source Separation** (VQSS) benefits from strong conditioning priors: the audio mixture already contains much of the target source, and the accompanying video provides frame-level constraints. As a result, many inference steps are unnecessary to achieve high-quality separation. In our experiments, using only 25 sampling steps yields an excellent trade-off between quality and efficiency—achieving $T_{A-V} = 96.88$ on VGGSOUND-Clean and 95.76 on VS-Hard, while running at 2.17 FPS (approximately $3\times$ faster than the 100-step setting, which runs at 0.72 FPS). For more real-time scenarios, even 10 steps suffice to maintain strong alignment (e.g., $T_{A-V} = 92.71$ on VGGSOUND-Clean and 94.07 on VS-Hard), achieving 4.00 FPS. Beyond 25 steps, the gains in quality are marginal, while throughput degrades significantly (e.g., 1.35 FPS at 50 steps, 0.72 FPS at 100). We conclude that 25 steps strike

a practical balance between separation quality and inference efficiency, making it a strong candidate for deployment in real-world applications.

### 5.6 QUALITATIVE RESULTS

To provide a more intuitive illustration of AlignSep's separation capability, we present qualitative comparisons with different VQSS methods in Figure 4. As shown in Figure 4a, when temporal cues between audio and video are misaligned, conventional semantics-based separation methods often fail to suppress irrelevant sounds. For example, although the drumming action in the video has already stopped, OmniSep still produces drum sounds (red regions). In contrast, AlignSep effectively leverages cross-modal correspondence and avoids generating spurious signals (green regions), separating audio strictly in accordance with the drumming rhythm. In Figure 4b, we highlight the spectral holes artifact, a common issue in mask-based separation methods when handling overlapping signals. While traditional approaches struggle to preserve continuity under such conditions (red regions), AlignSep mitigates these artifacts through its generative framework, yielding more complete and natural separation results. Additional qualitative examples are provided on our project page `https://AlignSep.github.io`.

## 6 CONCLUSION

In this work, we introduced AlignSep, the first generative framework for Video Query Sound Separation (VQSS) based on flow matching. By integrating temporal consistency mechanisms and a generative separation paradigm, AlignSep effectively addresses key challenges such as spectral holes, hallucination artifacts, and incomplete alignment. Our in-depth analysis reveals the fundamental differences between traditional single-conditioned flow matching tasks (e.g., text-to-audio) and the multi-conditioned generation setting of VQSS. To facilitate rigorous evaluation, we also constructed a challenging benchmark, VGGSound-Hard, composed of samples with homogeneous interference where temporal grounding is critical for successful separation. Extensive experiments demonstrate that AlignSep achieves state-of-the-art performance both quantitatively and perceptually across a range of benchmarks, validating its robustness and practical applicability.

### ETHICAL CONSIDERATIONS

Our work focuses on the development of a generative audiovisual source separation system, which aims to improve perceptual quality and alignment between sound and vision. While this technology has potential applications in video editing, accessibility, and content understanding, we acknowledge potential misuse such as manipulation or deepfake-style content generation. To mitigate such risks, we do not train or release models for identity synthesis or cross-modal generation beyond the scope of separation. All training data used in this work (MUSIC, VGGSound, and our VGGSound-Hard subset) are publicly available datasets collected for research purposes. No personally identifiable information (PII) is included, and we ensure compliance with the licenses and usage guidelines of each dataset. Human evaluations (e.g., MOS) were conducted anonymously with informed consent. We encourage responsible usage of this technology and explicitly discourage applications involving surveillance, impersonation, or deceptive audio generation.

### REPRODUCIBILITY STATEMENT

All code, pretrained models, and related resources will be publicly released upon paper acceptance under a permissive license, to encourage further research and community adoption. Detailed implementation settings and training protocols are thoroughly documented in the paper to facilitate reproducibility and independent verification.

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

## A  IMPLEMENTATION DETAILS

Table 4: Architecture details of 1D VAE for spectrogram compression.

| Hyperparameter | 1D VAE |
|---|---|
| Input tensor shape for 8-sec audio | (80,512) |
| Embedding dimension | 20 |
| Channels | 224 |
| Channel multiplier | 1, 2, 4 |
| Downsample layer position | after block 1 |
| Attention layer position | after block 3 |
| Output tensor shape for 8-sec audio | (20,256) |

The detailed hyperparameters for the VAE are provided in Table 4. This one-dimensional convolutional VAE model (1D VAE) is designed to process 8-second audio clips. For an input tensor with a shape of (80, 512), the model encodes it into a compact latent representation with an embedding dimension of 20, resulting in an output tensor with a shape of (20, 256). Regarding the network architecture, the initial number of channels is set to 224, and channel multipliers of 1, 2, and 4 are used to increase the channel depth in subsequent layers. To efficiently learn a hierarchy of features, the model applies a downsampling operation after the first convolutional block and integrates an attention mechanism after the third block to better capture important acoustic features.

Table 5: Hyperparameters of the vector field estimator of AlignSep.

| Hyperparameter | AlignSep |
|---|---|
| Layers | 4 |
| Hidden dimension | 576 |
| Attention heads | 8 |
| Conv1D-FFN dimension | 2,304 |
| Number of parameters | 158.94M |

The vector field estimator of AlignSep is given in Table 5. It is designed as a four-layer architecture with a hidden dimension of 576, eight attention heads, and a Conv1D-based feed-forward network (FFN) with an intermediate dimension of 2,304, totaling approximately 158.94M parameters. Notably, the hidden size is chosen to align with the feature representations: the generated audio is represented with 288 dimensions, while both the video and the reference audio are represented with 144 dimensions. This configuration ensures that the model effectively captures and enriches the details of the generated audio by integrating complementary cues from the visual and reference audio modalities. Moreover, we concatenate all information together to achieve better temporal consistency.

## B  MOS EVALUATION

We conducted a Mean Opinion Score (MOS) evaluation to assess the perceptual quality of the generated audio across four dimensions: Noise Residuals, Audio-Visual Consistency, Audio Quality, and Overall Score. We display the rating criteria of MOS in Table 7.

- Participants: 3 proficient annotator were recruited, with diverse backgrounds to reduce bias. Each participant was compensated for their time.
- Samples: For each method and each subset, we randomly selected 100 audio clips. Each participant was presented with the same randomized set of samples to ensure consistency.
- Scoring Protocol: Listeners rated each sample on a 5-point Likert scale (1 = Very Poor, 5 = Excellent) according to the four predefined dimensions (see Table 2). Detailed definitions of each dimension were provided beforehand to calibrate annotators' understanding.
- Randomization: The order of both methods and samples was randomized per participant to avoid ordering effects or bias toward specific systems.
- Aggregation: For each dimension, we report the mean and standard deviation across all raters and samples. Scores are averaged first across raters for each sample, and then across samples to obtain the final MOS.

Table 6: Mean Opinion Score (MOS) Rating Criteria

| Dimension | Score | Description |
|---|---|---|
| Noise Residuals | 1 | **Very Noisy:** Strong background noise that significantly affects intelligibility. |
| | 2 | **Noisy:** Noticeable noise, but speech or content remains intelligible. |
| | 3 | **Acceptable:** Minor noise present, generally tolerable. |
| | 4 | **Clean:** Little to no residual noise; very mild artifacts may exist. |
| | 5 | **Very Clean:** Completely free of noise or artifacts. |
| Audio-Visual Consistency | 1 | **Inconsistent:** Completely misaligned with visual content or scene context. |
| | 2 | **Low Consistency:** Partially related, but mostly inconsistent. |
| | 3 | **Moderate:** Generally acceptable but with clear mismatches. |
| | 4 | **Consistent:** Mostly aligned with minor inconsistencies. |
| | 5 | **Perfectly Consistent:** Fully synchronized and semantically coherent with visual content. |
| Audio Quality | 1 | **Very Poor:** Severely distorted, broken, or unnatural audio. |
| | 2 | **Poor:** Audible artifacts and degraded quality. |
| | 3 | **Fair:** Intelligible but lacks clarity or sounds slightly artificial. |
| | 4 | **Good:** Clear and natural with minor imperfections. |
| | 5 | **Excellent:** Highly natural, smooth, and human-like audio quality. |
| Overall Score | 1 | **Very Poor:** Unacceptable overall experience. |
| | 2 | **Poor:** Noticeable flaws that degrade the overall experience. |
| | 3 | **Fair:** Usable but with evident limitations. |
| | 4 | **Good:** Overall pleasant and functional. |
| | 5 | **Excellent:** High-quality and highly realistic audio experience. |

## C  USE OF LLM.

We used a large language model (LLM) solely for polishing the language of this paper.

## D  MORE EXPERIMENTS

### D.1  ABLATION STUDY ON GENERATIVE MODEL CHOICE.

As shown in table 7, on VGGSound-Clean, which demands higher semantic understanding, replacing Flow-Matching with diffusion leads to a modest performance drop, indicating that Flow-Matching helps improve the model's upper bound. On VGGSound-Hard, the performance drop of

Table 7: Ablation study on generative model choice.

| Method | VS-Clean ($S_{A-A}$) | VS-Clean ($S_{A-V}$) | VS-Clean ($T_{A-V}$) | VS-Hard ($T_{A-V}$) |
|---|---|---|---|---|
| AlignSep | **73.38** | **27.89** | **96.88** | **95.76** |
| AlignSep (w/o Flow-Matching) | 64.12 | 24.66 | 93.37 | 92.28 |
| AlignSep (w/o CAVP) | 69.21 | 26.76 | 94.71 | 76.27 |

Table 8: Ablation on different temporal fusion strategies.

| Method | VS-Clean ($S_{A-A}$) | VS-Clean ($S_{A-V}$) | VS-Clean ($T_{A-V}$) | VS-Hard ($T_{A-V}$) |
|---|---|---|---|---|
| cross-attention | **70.13** | **27.12** | **93.29** | **73.38** |
| concat | **73.38** | **27.89** | **96.88** | **95.76** |

AlignSep without Flow-Matching is minor (3.48), whereas removing CAVP causes a significant drop (19.49). This demonstrates that the temporal consistency in AlignSep relies primarily on effective visual temporal understanding rather than just the generative modeling method.

## D.2 ABLATION ON DIFFERENT TEMPORAL FUSION STRATEGIES.

Temporal fusion strategy is a key factor in our task. To evaluate its impact, we compared cross-attention with our concat fusion method in table 8. Results show that the two perform similarly on VS-clean (semantic-oriented), but on VS-hard—where strict temporal alignment is required—cross-attention fails almost entirely to capture temporal correspondence. This highlights the advantage of our concat-based fusion for temporal modeling.

