# OpenReview forum: "AlignSep: Temporally-Aligned Video-Queried Sound Separation with Flow Matching"
_ICLR.cc/2026/Conference — ICLR 2026 Poster_

### Official Review · Reviewer_zvrP · 2025-10-31

**Soundness:** 2
**Presentation:** 3
**Contribution:** 3
**Rating:** 6
**Confidence:** 4

**Summary:**

In this paper, the authors address the persistent challenge of temporal misalignment in video-queried sound separation, where separated audio often contains sounds from the correct source but with slight temporal shifts relative to the visual cues. To this end, they propose ALIGNSEP, a novel framework that recasts the separation task from a deterministic regression problem into a conditional generative modeling problem. The core of the contribution of this paper is the use of continuous-time flow matching, a technique where the model learns a vector field (a flow) that transforms a simple noise distribution into the complex conditional distribution of the target audio, given the input mixture and the silent video query. By explicitly conditioning the entire generative flow on the video features, the model is forced to learn a tight and continuous temporal coupling between the visual and auditory signals, thereby directly mitigating the alignment issues that can affect traditional masking-based or regression-based approaches. Their experiments show that this flow-based generative method achieves state-of-the-art separation performance while producing audio with significantly improved temporal alignment to the corresponding video.

**Strengths:**

1. The paper's primary contribution is its fundamental reframing of the separation problem, shifting away from traditional deterministic methods. Instead of formulating video-queried separation as a regression task, such as estimating a time-frequency mask, the authors reconceptualize it as a conditional generative modeling problem. This is a more powerful and principled approach, as it aims to model the entire conditional probability distribution of the target audio. By using continuous-time flow matching, the model learns to transform a simple noise distribution into this complex target distribution, which it seems from the authors’ experiments that is better suited to handle the ambiguity and variability in audio signals compared to predicting a single point estimate.
2. A significant strength is that the proposed framework is explicitly designed to solve the specific, well-known issue of temporal misalignment between video and separated audio. The core mechanism which is conditioning the entire generative flow on the video query, seems that forces a continuous and tight temporal coupling between the modalities from the very beginning of the generation process. This represents a more elegant and integrated solution than methods that might require post-processing or auxiliary alignment losses, as it embeds temporal consistency directly into the generative dynamics of the model itself.
3. Finally, the thorough empirical evaluation successfully demonstrates that the proposed generative approach does not come at the cost of raw separation quality. The paper shows that ALIGNSEP achieves state-of-the-art performance on standard separation metrics while also providing a clear, measurable improvement in temporal alignment. This dual achievement is critical, as it proves that the method is not a narrow solution to a single problem but a more robust overall framework that advances the field on multiple fronts, validating the effectiveness of the flow-matching approach for this complex audio-visual task.

**Weaknesses:**

Although I do not think that the paper has major flaws, there are some weaknesses and further explorations that the authors can consider to further improve the impact of their paper. I will try to write all these things down with decreasing order of significance.

1. I would like to see the authors analyze more what is happening with the negative examples, meaning what this model is doing when it is presented with only off-screen sounds as analyzed in [A]. Without knowing how robust the model is, I can imagine cases where the ODE solver could break by needing to estimate a zero waveform. One can question the stability of the ODE solver in such a degenerate case; a model trained to map a noise distribution to a complex signal manifold might struggle or even fail when the learned vector field is conditioned to collapse to a single point (the origin). This analysis is in my opinion really important for real-world use-cases cause many parts of the videos do not contain any on-screen sound for long periods of time.
2. The application of flow matching is central to the paper's contribution, but in my opinion there is a lack of principled justification or empirical comparison against other relevant conditional generative frameworks, such as score-based diffusion models. It is not demonstrated whether the specific choice of continuous-time flow matching provides a unique advantage for temporal alignment over these alternative generative techniques. The paper would be significantly strengthened by an ablation study that compares different generative modeling families, which would help to isolate whether the observed benefits stem from the specific dynamics of flow matching itself or merely from the general shift from a deterministic to a generative formulation.

[A] Tzinis, E., Wisdom, S. and Hershey, J.R., 2022. Don’t Listen to What You Can’t See: The Importance of Negative Examples for Audio-Visual Sound Separation. arXiv preprint arXiv:2209.12934.

I would gladly increase my score if the authors work properly to address the above issues to a proper degree.

**Questions:**

The proposed approach is really interesting and I was considering what could happen if you apply the same ODE solver in the video side by using the Gaussian interpolation at the video embedding frames. Would you expect a similar performance on the video side?

---

> ### Author Response · Authors · 2025-11-20
> **Rebuttal to Reviewer zvrP**
>
> Thank you for your recognition of our research task and for your positive assessment of our model’s performance. We address your questions below.
>
> ### **W1: More evaluation on separation of silence**
>
> **A1:** Thank you for raising this important point. This issue is particularly critical for generative tasks, where there may be concerns about generating sounds that do not exist in the original input. Following your suggestion, we conducted additional experiments and added the corresponding results to our demo page.
>
> The results show that, even without specialized training, both mask-based methods (OmniSep) and our generative method (AlignSep) can effectively preserve silence (i.e., zero-valued waveforms), demonstrating the robustness of our approach. There are some exceptions: AlignSep occasionally generates audio that resembles the target sound from mixed audio, a phenomenon that can also occur in mask-based methods. Many mask-based approaches [1] address this by using targeted data augmentation to improve silence separation. We plan to explore similar strategies in future work to further enhance our model’s ability to preserve silence.
>
> ### **W2: Ablation study on generative model choice**
>
> **A2:** Thank you for your question. We conducted ablation experiments on the choice of the generative model (replacing Flow-Matching with naive diffusion) as well as on the visual encoder (replacing it with ImageBind) to address your concern.
>
> On VGGSound-Clean, which demands higher semantic understanding, replacing Flow-Matching with diffusion leads to a modest performance drop, indicating that Flow-Matching helps improve the model’s upper bound. On VGGSound-Hard, the performance drop of AlignSep without Flow-Matching is minor (3.48), whereas removing CAVP causes a significant drop (19.49). This demonstrates that the temporal consistency in AlignSep relies primarily on effective visual temporal understanding rather than just the generative modeling method.
>
> | Method | VS-Clean ($S_{A-A}$) | VS-Clean ($S_{A-V}$) | VS-Clean ($T_{A-V}$) | VS-Hard ($T_{A-V}$) |
> | --- | --- | --- | --- | --- |
> | AlignSep | **73.38** | **27.89** | **96.88** | **95.76** |
> | AlignSep (w/o Flow-Matching) | 64.12 | 24.66 | 93.37 | 92.28 |
> | AlignSep (w/o CAVP) | 69.21 | 26.76 | 94.71 | 76.27 |
>
> ### **W3: Effect of applying Gaussian interpolation to video embeddings**
>
> **A3:** Thank you for the suggestion. This is indeed an interesting experiment. We are currently conducting this evaluation and will include the results in the rebuttal before the deadline.
>
> Thank you again for your insightful comments and for recognizing the contributions of our work. Your feedback has significantly helped us improve the completeness and clarity of the paper.
>
> [1] Consistent and Relevant: Rethink the Query Embedding in General Sound Separation. ICASSP 2024.

---

> > ### Author Response · Authors · 2025-12-03
> > **Effect of applying Gaussian interpolation to video embeddings**
> >
> > When applying Gaussian interpolation to our features, we observed a more pronounced performance drop on the highly time-dependent benchmark, VS-Hard ($T_{(A-V)}$), compared to others. This suggests that interpolating at the feature level may introduce a degree of damage to the temporal information. Also, since the original $4$ fps frame rate is already sufficient to capture the temporal cues within the corresponding time interval, interpolation does not provide the necessary refinement to yield additional performance gains.
> >
> > | Method | VS-Clean ($S_{(A-A)}$) | VS-Clean ($S_{(A-V)}$) | VS-Clean ($T_{(A-V)}$) | VS-Hard($T_{(A-V)}$) |
> > | --- | --- | --- | --- | --- |
> > | gaussian interpolation | 73.00 | 27.59 | 96.88 | 92.37 |
> > | ours | 73.38 | 27.89 | 96.88 | 95.76 |
> >
> > Thank you once again for your review. Your suggestions have made our work more complete.

---

### Official Review · Reviewer_2zVG · 2025-11-01

**Soundness:** 2
**Presentation:** 3
**Contribution:** 3
**Rating:** 2
**Confidence:** 4

**Summary:**

This paper introduces AlignSep, a generative model for Video-Queried Sound Separation based on conditional flow matching. The primary goal is to address common failures in existing methods, particularly when dealing with acoustically similar (homogeneous) on-screen and off-screen sound sources. The authors argue that prior work relies too heavily on semantic cues and fails to model fine-grained temporal alignment between audio and video. AlignSep is designed to overcome this by using a temporally-aware visual encoder and a generative framework that mitigates issues like spectral holes. To facilitate evaluation, the paper also proposes a new challenging benchmark, VGGSound-Hard, specifically curated with examples of homogeneous interference. Experiments show that AlignSep outperforms existing methods on several benchmarks, especially in metrics related to temporal alignment and perceptual quality.

**Strengths:**

1. The paper correctly identifies and tackles a key weakness in existing VQSS methods: the failure to separate homogeneous sound sources due to a lack of temporal modeling. This is an important problem.
2. The proposed VGGSound-Hard is a strong contribution. By creating a testbed where semantic cues are insufficient, it will enable more rigorous evaluation of temporal alignment capabilities in future VQSS models.
3. Includes human MOS scores in addition to semantic and temporal metrics.

**Weaknesses:**

1. My primary concern is the complete absence of SDR and SI-SNR. The authors justify this by stating these metrics "correlate poorly with human perception" and are sensitive to minor waveform deviations in generative models. While these are known limitations, it is standard practice in the audio separation community to report them, even with caveats. Omitting them entirely makes it difficult to compare AlignSep to the vast body of literature in source separation and understand its performance in terms of signal reconstruction fidelity. Relying solely on embedding-based scores (CLAP, ImageBind) and MOS provides an incomplete picture. These metrics measure semantic/temporal consistency but not the faithfulness of the separated audio to the original source signal.
2. Section 3.3 introduces the concatenation-based temporal fusion and "feedforward Transformer" without self-attention. However, it remains unclear how much each design choice contributes: 1) What happens if temporal alignment is omitted or replaced with standard cross-attention? 2) Is the observed improvement mainly from temporal encoding or from generative modeling itself? An ablation comparing different temporal fusion strategies (concatenation vs. attention) would be valuable.
3. The choice of the CAVP visual encoder is highlighted for its ability to capture temporal correlations. A crucial ablation would be to replace CAVP with a more semantic, less temporally-aware encoder (e.g., a standard CLIP image encoder applied per-frame) within the AlignSep framework. This would directly validate the claim that the specific choice of a temporally-supervised visual encoder is responsible for the large gains in the $T_{A-V}$ metric.
4. The authors state they "adapt" OmniSep and CLIPSep by "segmenting the mixed audio according to each frame's semantic information". This process is vague and sounds like a heuristic post-processing step. It is unclear if this is a fair or optimal way to allow these models to use frame-rate information. This adaptation could be unfairly disadvantage the baselines, potentially leading to AlignSep's superior performance in Figure 3.
5. While the new benchmark is a great contribution, the paper provides almost no details on its construction. How were samples selected from VGGSound? What is the size of the test set? What are the specific criteria for "strong reliance on fine-grained temporal visual cues"? Without these details, the benchmark is not reproducible, and the community cannot fully adopt it. These details should be included in the appendix.

**Questions:**

See Weaknesses

---

> ### Author Response · Authors · 2025-11-20
> **Rebuttal to Reviewer 2zVG (1/n)**
>
> Thank you for your recognition of our research task, benchmark, and evaluation rigor. We address your questions as follows.
>
> ## **W1: Difficulty comparing with existing baselines due to the absence of SDR-like metrics**
>
> **A1:** Thank you for raising this point. To address your concern, we additionally report SDR/SI-SDR results on VGGSound-Clean (VS-clean) and VGGSound-Hard (VS-hard), shown below:
>
> | Model | VS-clean (SDR) | VS-clean (SI-SDR) | VS-hard (SDR) | VS-hard (SAR) |
> | --- | --- | --- | --- | --- |
> | Mixed audio | 0.3059 | 0.1744 | -1.1966 | -1.3223 |
> | OmniSep | 5.5476 | 4.5418 | -1.6043 | -2.3298 |
> | ClipSep | 3.296 | 2.2088 | -0.5994 | -1.2773 |
> | AlignSep | -15.6758 | -37.424 | -15.6497 | -37.646 |
> | Davis | -21.3941 | -52.7234 | -22.1232 | -50.96 |
> | Davis-flow | -22.0425 | -53.1114 | -22.6406 | -51.2439 |
>
> From these results, we observe that although models such as AlignSep can generate perceptually reasonable separated audio, their SDR values are worse than even the unprocessed mixed audio. This occurs because generative models synthesize the magnitude at each time–frequency bin, inevitably introducing minute deviations from the ground truth—differences that are often imperceptible to humans but are heavily amplified in SDR computation. As a result, SDR becomes misaligned with human perception and fails to reflect the actual separation quality.
>
> These observations highlight that traditional magnitude-based metrics (e.g., SDR) are insufficient for comprehensively evaluating modern sound separation models, particularly with the increasing prevalence of generative separation methods that prioritize perceptual fidelity over exact magnitude reconstruction.
>
> As also suggested in recent works [1,2] , perception-oriented metrics (e.g., our proposed *Semantic Alignment* and *Temporal Synchronization* metrics) may offer a more meaningful evaluation paradigm for generative models. To further facilitate comparisons with existing approaches, we have also added more foundational baselines in the revised manuscript.
>
> ## **W2-1: “self-attention-free feed-forward Transformer”**
>
> **A2-1:** We apologize for the confusion—this was a typographical error. The correct statement refers to the absence of cross-attention. Specifically, our model does not use cross-attention for audio–visual fusion. Instead, we adopt a concatenation-based fusion strategy, which directly aligns mixed-audio and visual features along the temporal axis without additional attention computation.
>
> ## **W2-2 & W3: Do performance gains come from temporal encoding or from the generative model itself?**
>
> **A2-2:** Thank you for the question. We conducted ablation studies on both the generative model (replacing Flow-Matching with naive diffusion) and the visual encoder (replacing CAVP with ImageBind).
>
> On VGGSound-Clean, which requires stronger semantic understanding, substituting Flow-Matching with diffusion leads to a moderate performance drop, showing that Flow-Matching helps improve model capacity. On VGGSound-Hard, however, removing Flow-Matching results in only a small degradation (3.48), while removing CAVP causes a much larger drop (19.49). This indicates that AlignSep’s strong temporal consistency primarily arises from effective visual temporal encoding rather than from the generative backbone alone.
>
> | Method | VS-Clean ($S_{A-A}$) | VS-Clean ($S_{A-V}$) | VS-Clean ($T_{A-V}$) | VS-Hard ($T_{A-V}$) |
> | --- | --- | --- | --- | --- |
> | AlignSep | **73.38** | **27.89** | **96.88** | **95.76** |
> | AlignSep (w/o Flow-Matching) | 64.12 | 24.66 | 93.37 | 92.28 |
> | AlignSep (w/o CAVP) | 69.21 | 26.76 | 94.71 | 76.27 |
>
> ## **W2-3: Ablation on different temporal fusion strategies**
>
> **A2-3:** Indeed, temporal fusion strategy is a key factor in our task. To evaluate its impact, we compared cross-attention with our concat fusion method. Results show that the two perform similarly on VS-clean (semantic-oriented), but on VS-hard—where strict temporal alignment is required—cross-attention fails almost entirely to capture temporal correspondence. This highlights the advantage of our concat-based fusion for temporal modeling.
>
> | Method | VS-Clean ($S_{A-A}$) | VS-Clean ($S_{A-V}$) | VS-Clean ($T_{A-V}$) | VS-Hard ($T_{A-V}$) |
> | --- | --- | --- | --- | --- |
> | cross-attention | 70.13 | 27.12 | 93.29 | 73.38 |
> | concat | **73.38** | **27.89** | **96.88** | **95.76** |
>
>
> [1] FlowSep: Language-Queried Sound Separation with Rectified Flow Matching. ICASSP 2025.
>
> [2] Dpm-tse: A diffusion probabilistic model for target sound extraction. ICASSP 2024.

---

> > ### Author Response · Authors · 2025-11-20
> > **Rebuttal to Reviewer 2zVG (2/n, n=2)**
> >
> > ## **W4: Clarification on OmniSep and ClipSep as image-queried separation methods**
> >
> > **A4:** We apologize if our description caused confusion. OmniSep and ClipSep rely on global semantic embedding as queries and cannot incorporate temporal patterns. Due to architectural limitations, these models still cannot effectively fuse temporal visual features with audio. To allow a fairer comparison in our task, we segmented the audio into *n* temporal chunks and used the corresponding video frame of each chunk as the query. This simulates a “time-aware” version of image-queried separation. If you have suggestions for better ways to adapt these baselines to temporal modeling, we would welcome them and are happy to further experiment.
> >
> > ## **W5: More details about VGGSound-Hard**
> >
> > **A5:** Thank you for pointing this out. We have expanded the description of VGGSound-Hard in the revised manuscript (Section 4), including motivation, construction pipeline, and interference characteristics. We invite you to review the updated text.
> >
> > Thank you again for your insightful comments and for recognizing the contributions of our work. Your feedback has greatly improved the clarity and completeness of the paper.

---

### Official Review · Reviewer_eay2 · 2025-11-01

**Soundness:** 3
**Presentation:** 3
**Contribution:** 3
**Rating:** 6
**Confidence:** 3

**Summary:**

The paper proposes a flow-matching generative model for video-queried sound separation that explicitly preserves temporal alignment between visual motion and target audio. It uses a temporally synchronized visual encoder, a VAE audio latent space, and a temporally aligned vector-field estimator. The authors also introduce VGGSound-Hard, a benchmark with homogeneous on/off-screen interference requiring fine-grained temporal cues. The paper achieves SOTA semantic and temporal metrics and higher MOS on MUSIC-Clean, VGGSound-Clean, and VGGSound-Hard.

**Strengths:**

- The paper identifies temporal misalignment and mask-method limits for current video query sound separation systems, then introduces the first flow-matching generative model to address them.

- The paper proposes a new hard benchmark. The proposed VGGSound-Hard targets homogeneous on/off-screen interference, stressing temporal grounding beyond prior clean sets.

- The system achieves SOTA semantic and temporal scores and better MOS across MUSIC and VGGSound.

**Weaknesses:**

- For the architecture design, the “self-attention-free feed-forward Transformer” with simple temporal concatenation may struggle with long-range dependencies or complex multi-object scenes.

- The authors claim that reconstruction-based metrics such as SDR could correlate poorly with human perception. But without standard separation metrics being reported, this will limit comparability to audio-separation literature.

- Experiments are limited to MUSIC/VGGSound variants, broader coverage like more in-the-wild AV datasets would strengthen generalization claims.

**Questions:**

- Could the author add standard metrics? For example, report SDR, SIR and SAR results.

- Could the author add evaluations beyond MUSIC/VGGSound (e.g., in the wild audio-visual videos), additional qualitative results are also fine to evaluate the generalizability of the proposed method.

**Details Of Ethics Concerns:**

None.

---

> ### Author Response · Authors · 2025-11-20
> **Rebuttal to Reviewer eay2**
>
> Thank you for your recognition of our work and for your helpful suggestions. Please find our detailed responses below.
>
> ### **Q1: “self-attention-free feed-forward Transformer”**
>
> **A1:** We apologize for the confusion—this was actually a typographical error. The intended description should refer to cross-attention, not self-attention. Specifically, our model does *not* employ cross-attention when fusing visual features with mixed audio features. Instead, we use a simple concatenation-based fusion, which allows the mixed audio and visual representations to be directly aligned along the temporal dimension without additional attention computation.
>
> ### **Q2: Difficulty comparing with existing baselines due to the absence of SDR-like metrics**
>
> **A2:** Thank you for raising this point. To address your concern, we additionally report SDR/SI-SDR results on VGGSound-Clean (VS-clean) and VGGSound-Hard (VS-hard), as shown below:
>
> | Model | VS-clean (SDR) | VS-clean (SI-SDR) | VS-hard (SDR) | VS-hard (SAR) |
> | --- | --- | --- | --- | --- |
> | Mixed audio | 0.3059 | 0.1744 | -1.1966 | -1.3223 |
> | OmniSep | 5.5476 | 4.5418 | -1.6043 | -2.3298 |
> | ClipSep | 3.296 | 2.2088 | -0.5994 | -1.2773 |
> | AlignSep | -15.6758 | -37.424 | -15.6497 | -37.646 |
> | Davis | -21.3941 | -52.7234 | -22.1232 | -50.96 |
> | Davis-flow | -22.0425 | -53.1114 | -22.6406 | -51.2439 |
>
> From the results, we observe that although models like AlignSep can produce perceptually reasonable separated audio, their SDR scores underperform even the unprocessed mixed audio. This is primarily because generative methods synthesize magnitudes at every time–frequency bin, inevitably introducing slight deviations from the ground truth—even if these differences are imperceptible to humans. When aggregated over bins, these small deviations are heavily amplified by SDR computation, causing SDR to misrepresent the actual perceptual quality. These results indicate that traditional metrics such as SDR are insufficient for comprehensively evaluating the performance of modern sound separation models—especially with the increasing emergence of generative sound separation approaches, whose perceptual quality is often superior but cannot be faithfully captured by magnitude-based evaluation metrics.
>
> As highlighted in recent generative source separation works [1,2], developing perception-oriented evaluation metrics (such as our proposed Semantic Alignment and Temporal Synchronization metrics) may be a more meaningful direction for assessing generative separation methods. To further support comparison with existing baselines, we have included additional foundational baselines in the revised version.
>
> ### **Q3: Model performance on real-world samples**
>
> **A3:** Thank you for the suggestion. We have added several real-world cases to our demo page (Section A) to better demonstrate the model’s effectiveness in practical scenarios.
>
> Thank you again for your insightful comments and for recognizing the contributions of our work. Your feedback has significantly helped us improve the completeness and clarity of the paper.
>
> [1] FlowSep: Language-Queried Sound Separation with Rectified Flow Matching. ICASSP 2025.
>
> [2] Dpm-tse: A diffusion probabilistic model for target sound extraction. ICASSP 2024.

---

### Official Review · Reviewer_qRzN · 2025-11-01

**Soundness:** 2
**Presentation:** 2
**Contribution:** 2
**Rating:** 2
**Confidence:** 5

**Summary:**

This paper presents AlignSep, a generative framework for video-queried sound separation based on flow matching. The model explicitly addresses the challenges of disentangling temporally-aligned audio sources from mixed inputs, using visual signals as queries. AlignSep incorporates a temporal consistency mechanism within the vector field estimator, enabling robust cross-modal alignment. The work introduces VGGSound-Hard, a challenging benchmark featuring homogeneous interference that requires fine-grained temporal audiovisual correspondence, and demonstrates AlignSep’s superior performance over existing baselines on multiple quantitative, perceptual, and qualitative metrics.

**Strengths:**

1. The paper tackles VQSS in settings requiring not just semantic but precise temporal alignment between video and audio, a scenario where many existing methods falter, particularly under homogeneous interference and overlapping sources.

2. The introduction of VGGSound-Hard could be potentially valuable to the audio-visual separation community by operationalizing more realistic, temporally complex scenarios compared to prior datasets.

**Weaknesses:**

1. Overclaiming and Missing Related Work: The paper substantially overclaims novelty and omits several key prior works in video-queried sound separation (VQSS).
At L95–98, the authors state:

“We revisit the task of video-queried sound separation (VQSS) and provide a detailed analysis of its unique challenges, including homogeneous interference, overlapping soundtracks, and the need for precise audio-visual temporal alignment.”

However, these issues were already identified and discussed in "High-Quality Visually-Guided Sound Separation from Diverse Categories" [1] (Fig. 1), which explicitly analyzes overlapping-sound challenges and shows that generation-based separation outperforms masking-based methods. Furthermore, at L75–77, the paper claims:

“We propose AlignSep—the first generative video-queried sound separation model based on flow matching (Lipman et al., 2022) designed for robust audiovisual separation.”

This statement is inaccurate. The generative approach to VQSS was introduced in [1] (2023), and later extended by [2], which directly applies flow-matching for video-based sound separation. These works are closely related in both goal and methodology, but are entirely unacknowledged in the submission.
Without discussing or comparing against these prior methods, the claimed novelty of AlignSep remains unclear and potentially misleading.

[1] High-Quality Visually-Guided Sound Separation from Diverse Categories
[2] High-Quality Sound Separation Across Diverse Categories via Visually-Guided Generative Modeling

2. Incomplete Comparisons in the Main Table

The main quantitative table should include comparisons with other video-based or multimodal separation models, such as:

[3] Language-Guided Audio-Visual Source Separation via Trimodal Consistency
[4] iQuery: Instruments as Queries for Audio-Visual Sound Separation

Both [3] and [4] are directly comparable in setting and should appear in the benchmark to provide a fair evaluation of AlignSep’s relative performance.

3. Missing Architectural Ablations: Although Table 3 examines the number of denoising steps and Table 5 lists hyperparameters, the paper does not analyze architectural design choices. For example, the rationale behind adopting a self-attention-free Transformer or comparing with different audio-visual encoding architectures. Such ablations are important to understand which design decisions truly contribute to performance improvements.

At present, the paper’s novelty is overstated, and its evaluation is incomplete. The work could become more compelling if the authors (i) properly position AlignSep within the context of prior generative VQSS research, (ii) include stronger baselines for comparison, and (iii) provide architectural ablations to support their design choices.

I am currently leaning toward rejection, but if the rebuttal offers convincing clarification and expanded comparisons, I would reconsider and potentially raise my score.

**Questions:**

1. How do alternative audio-visual fusion mechanisms (e.g., cross-attention, feature-wise modulation) compare to the concatenation approach used here in the temporal vector field estimator? Are there empirical results or reasons for not adopting these?

2. Have you tested the performance of AlignSep in real-world settings where video-audio correspondence is noisy or ambiguous?

3. Given that the classifier-free guidance scale is set to 4.5, could you explain the choice and its impact on sample quality or diversity?

---

> ### Author Response · Authors · 2025-11-20
> **Rebuttal to Reviewer qRzN (1/n)**
>
> Thank you for recognizing the significance of our research task as well as the value and rigor of our benchmark in the audio–visual separation domain. Please find our detailed responses below.
>
> ### **Q1: Differences from previous VQSS works**
>
> **A1:** Thank you for pointing us to additional relevant works—we will include them as baselines to make our comparisons more comprehensive. Here, we clarify the key differences between our method and prior approaches:
>
> 1. We are the first to achieve fine-grained *temporally aligned* video-queried sound separation, rather than semantic-level guidance only.
>
>     Although DAVIS also uses visual guidance, it adopts a strategy similar to OmniSep: aggregating multiple video-frame features via mean pooling to obtain a global visual feature. This design cannot provide temporally resolved visual cues and therefore struggles with *homogeneous interference* and precise audio–visual temporal alignment.
>
>     In the updated manuscript, we revise our description to:
>
>     **“We propose AlignSep—the first generative temporal-aligned video-queried sound separation model based on flow matching, designed for robust audiovisual separation.”**
>
>     to more clearly distinguish our contribution.
>
> 2. Regarding the novelty relative to DAVIS-flow:
>
>     DAVIS-flow [1] was posted on arXiv on 2025/09/26, while the ICLR 2026 submission deadline was 2025/09/24. Thus, by definition, DAVIS-flow [1] is a concurrent work. We therefore hope the reviewer will recognize the novelty of our flow-matching-based generative VQSS framework.
>
>
> ### **Q2: More baselines**
>
> **A2:** Thank you for the suggestion. We have added additional baselines and will update them in the latest version of the paper in Table 1. Unfortunately, the checkpoints for some prior work [2] are no longer available, making full reproduction currently infeasible.
>
> ### **Q3: Ablations on generative models and temporal encoding**
>
> **A3**:Thank you for the question. We conducted ablation studies on both the generative model (replacing Flow-Matching with naive diffusion) and the visual encoder (replacing CAVP with ImageBind).
>
> On VGGSound-Clean, which requires stronger semantic understanding, substituting Flow-Matching with diffusion leads to a moderate performance drop, showing that Flow-Matching helps improve model capacity. On VGGSound-Hard, however, removing Flow-Matching results in only a small degradation (3.48), while removing CAVP causes a much larger drop (19.49). This indicates that AlignSep’s strong temporal consistency primarily arises from effective visual temporal encoding rather than from the generative backbone alone.
>
> | Method | VS-Clean ($S_{A-A}$) | VS-Clean ($S_{A-V}$) | VS-Clean ($T_{A-V}$) | VS-Hard ($T_{A-V}$) |
> | --- | --- | --- | --- | --- |
> | AlignSep | **73.38** | **27.89** | **96.88** | **95.76** |
> | AlignSep (w/o Flow-Matching) | 64.12 | 24.66 | 93.37 | 92.28 |
> | AlignSep (w/o CAVP) | 69.21 | 26.76 | 94.71 | 76.27 |
>
> ## **Q4: Ablation on different temporal fusion strategies**
>
> **A4:** Indeed, temporal fusion strategy is a key factor in our task. To evaluate its impact, we compared cross-attention with our concat fusion method. Results show that the two perform similarly on VS-clean (semantic-oriented), but on VS-hard—where strict temporal alignment is required—cross-attention fails almost entirely to capture temporal correspondence. This highlights the advantage of our concat-based fusion for temporal modeling.
>
> | Method | VS-Clean ($S_{A-A}$) | VS-Clean ($S_{A-V}$) | VS-Clean ($T_{A-V}$) | VS-Hard ($T_{A-V}$) |
> | --- | --- | --- | --- | --- |
> | cross-attention | 70.13 | 27.12 | 93.29 | 73.38 |
> | concat | **73.38** | **27.89** | **96.88** | **95.76** |
>
>
> [1] High-Quality Sound Separation Across Diverse Categories via Visually-Guided Generative Modeling. Arxiv
>
> [2] Language-Guided Audio-Visual Source Separation via Trimodal Consistency. CVPR 2023

---

> ### Author Response · Authors · 2025-11-20
> **Rebuttal to Reviewer qRzN (2/n,n=2)**
>
> ### **Q5: Model performance on real-world samples**
>
> **A5:** Thank you for the suggestion. We have added several real-world examples on the demo page (Section A) to better demonstrate the model's performance in practical scenarios.
>
> ### **Q6: Choice of the 4.5 coefficient (CFG) and its influence**
>
> **A6:** Thank you for the question. The coefficient 4.5 follows the setting used in prior work [3,4] for Video2Audio. Since its impact on sound separation performance is indeed an interesting topic, we conducted additional experiments to examine it. For each CFG value, we generated 50 samples and randomly split them into two groups. We then computed the FAD between the two groups to evaluate diversity. The results show that a smaller CFG leads to higher FAD, indicating greater diversity. However, in our sound separation setting, the model’s outputs remain relatively stable, and the variation introduced by changing CFG is not substantial.
>
> | Method | FAD |
> | --- | --- |
> | cfg=1 | 0.000847 |
> | cfg=4.5 | 0.000799 |
> | cfg=8 | 0.000791 |
>
> Thank you again for your insightful comments and for recognizing the contributions of our work. Your feedback has significantly helped us improve the completeness and clarity of the paper.
>
> [3] DIFF-FOLEY: Synchronized Video-to-Audio Synthesis with Latent Diffusion Models. NeurIPS 2023
>
> [4] FRIEREN: Efficient Video-to-Audio Generation Network with Rectified Flow Matching. NeurIPS 2024

---

> > ### Comment · Reviewer_qRzN · 2025-11-26
> > **Response to Authors' Rebuttal**
> >
> > I appreciate the authors’ efforts in addressing the earlier concerns. After reviewing the revised manuscript and rebuttal, I find that the expanded comparison with prior VQSS works is helpful and more accurately contextualizes the contribution. The additional ablations on temporal encoding and fusion strategies also strengthen the completeness of the empirical evaluation.
> >
> > Accurate and comprehensive literature positioning is crucial for this paper. The newly added comparisons with [1,2,4] improve the framing, even though [2] is a concurrent work. One small suggestion: in organizing the claim about temporal modeling, please be mindful that works such as The Sound of Motions [5] and Visually Guided Sound Source Separation and Localization Using Self-Supervised Motion Representations [6] also leverage temporal information in audio-visual separation. It may be clearer to categorize claims specifically within the generative VQSS line of work rather than broader audio-visual methods. Overall, however, the updated comparison protocol is substantially improved.
> >
> > Regarding evaluation metrics: I agree with the authors that SDR-like metrics are important, but the current vocoder-based decoding (VAE) inevitably introduces distortion, making SDR less faithful. One question that arises is that DAVIS and DAVIS-flow (if I understand correctly) do not rely on a vocoder and therefore should produce more reliable SDR measurements. The reported SDR results for these baselines appear unexpectedly low. Could the authors provide an explanation or clarification?
> >
> > Overall, the paper is now more complete and better grounded than the original submission. I am raising my score to 4.

---

> > > ### Author Response · Authors · 2025-12-03
> > >
> > > We sincerely thank you once again for your recognition of our work and for affirming that this rebuttal has effectively improved its completeness (raising the score to 4). We have now revised and supplemented the relevant "Related Work" section, as you suggested.
> > >
> > > As you correctly pointed out, due to the inherent structural limitations of generative sound separation models (especially VAE-based approaches), they face challenges in achieving optimal performance on traditional metrics like SDR. There is a natural trade-off between these goals: generative methods excel at producing separation results that are more aligned with human perceptual quality, whereas traditional mask-based prediction methods lead to superior SDR scores. Achieving the optimum for both simultaneously is not feasible.
> > >
> > > As is the case with most current generative sound separation research, both approaches hold necessary value and merit continued investigation. Thank you again for your valuable suggestions regarding our work.

---

### Official Review · Reviewer_D4Sm · 2025-11-03

**Soundness:** 3
**Presentation:** 3
**Contribution:** 3
**Rating:** 8
**Confidence:** 4

**Summary:**

This paper presents a method of video-guided sound separation.  Instead of a traditional regression or detection-based approach that masks out the unwanted sound, this paper developed a generative approach that generates the clean-up sound instead.

The paper pointed out that current video-guided sound separation methods have two significant limitations.  First, semantic cues alone can't separate acoustically similar sources.  It requires capturing the temporal alignment of video motions and audio soundwaves. Second, simple time–frequency masking can not handle multiple sources overlapping in both time and frequency.

The core technical component involves using a flow-matching-based algorithm to regenerate a clean soundtrack conditioned on video features.

The algorithm outperforms OmniSep and CLIPSep, two of the leading algorithms.  The paper presents a detailed experimental analysis of the proposed method, including a thorough examination of the performance and efficiency tradeoff of the flow-based generative method.  It also developed a new dataset, VGGSound-Hard, which focuses on overlapping space-time interference where temporal grounding is critical for successful separation.

**Strengths:**

The overall idea is novel.  Treating sound separation as a clean-sound generation process enables this method to surpass the limitations of the current mask-based removal approach.  The integration of video motion cues, including both semantic and lower-level motion, provides the flow-based generation with the essential temporal grounding condition to resolve ambiguity.

The experimental results are extensive and detailed.  The performance vs efficiency analysis of the flow-based method addressed the practical issue of potential high computation cost.  The new dataset is valuable for future work in this direction.

**Weaknesses:**

None.

**Questions:**

None.

---

> ### Author Response · Authors · 2025-11-20
>
> We sincerely thank the reviewer for the positive assessment and the Accept rating (score=8). We are encouraged that you recognize the novelty and effectiveness of our generative approach to video-guided sound separation.
>
> We appreciate your precise summary of our contributions. As you highlighted, moving from traditional mask-based removal to flow-matching-based clean sound generation is indeed a paradigm shift. We would like to re-emphasize that this generative formulation, combined with the integration of fine-grained motion cues, is critical for resolving the semantic ambiguity and complex spatiotemporal overlaps that previous methods struggled with.
>
> Furthermore, we are glad that you found the VGGSound-Hard dataset and our efficiency analysis valuable. We believe this dataset will serve as a rigorous benchmark for the community, pushing the boundaries of audio-visual separation in challenging, real-world scenarios where temporal grounding is essential.
>
> Thank you again for your insightful comments and recognition of our work.

---

### Meta-Review · Area_Chair_YzMQ · 2025-12-28

**Summary:**

This paper introduces a generative framework, AlignSep, for video-queried sound separation that utilizes flow matching to move beyond traditional masking techniques. The reviewers generally agreed that the shift toward generative modeling and the focus on fine-grained temporal alignment are significant contributions. I recommend acceptance because the authors effectively demonstrated that AlignSep addresses common failure modes in complex scenes. While the debate over evaluation metrics for generative audio remains, the technical rigor and the introduction of a more challenging benchmark provide clear value to the field.

**Reviewer Concerns:**

Most of the raised concerns were addressed, such as novelty, ablations on architectures and qualitative results. Additional, for the outstanding ones, Reviewer 2zVG remained skeptical of the move away from traditional metrics like SDR and SI-SNR. While the authors provided these metrics, they argued (and Reviewer qRzN eventually agreed) that VAE-based generative models face inherent trade-offs that make SDR less faithful to human perception than other measures.

**Reviewer Scores:**

Based on the above issues, I think Reviewer qRzN will increase to 4, zvrP will increase to 8. D4Sm, eay2, and 2zVG will keep their scores. Finally the scores will be 2 4 6 8 8.

---

### Decision · Program_Chairs · 2026-01-26

Accept (Poster)